# Development of American-Lineage Influenza H5N2 Reassortant Vaccine Viruses for Pandemic Preparedness

**DOI:** 10.3390/v11060543

**Published:** 2019-06-11

**Authors:** Po-Ling Chen, Alan Yung-Chih Hu, Chun-Yang Lin, Tsai-Chuan Weng, Chia-Chun Lai, Yu-Fen Tseng, Ming-Chu Cheng, Min-Yuan Chia, Wen-Chin Lin, Chia-Tsui Yeh, Ih-Jen Su, Min-Shi Lee

**Affiliations:** 1National Institution of Infectious Diseases and Vaccinology, National Health Research Institutes (NHRI), Zhunan, Miaoli 35053, Taiwan; letitia@nhri.org.tw (P.-L.C.); alanhu@nhri.org.tw (A.Y.-C.H.); grayingaries@outlook.com (C.-Y.L.); wtc@nhri.org.tw (T.-C.W.); laicc2@nhri.org.tw (C.-C.L.); yufents@gmail.com (Y.-F.T.); chiaminyuan@dragon.nchu.edu.tw (M.-Y.C.) suihjen0704@stust.edu.tw (I.-J.S.); 2Institute of Molecular and Cellular Biology, National Tsing Hua University, Hsinchu 30013, Taiwan; 3College of Life Science, National Tsing Hua University, Hsinchu 30013, Taiwan; 4Department of Veterinary Medicine, College of Veterinary Medicine, National Pingtung University of Science and Technology, Pingtung 91201, Taiwan; mccheng@mail.npust.edu.tw; 5Animal Health Research Institutes, Danshui, New Taipei City 25158, Taiwan; 6Department of Veterinary Medicine, National Chung Hsing University, Taichung 40227, Taiwan; 7Institute of Preventive Medicine, National Defence Medical Centre, Taipei 23742, Taiwan; spps057@gmail.com (W.-C.L.); yyhome@mail.ndmctsgh.edu.tw (C.-T.Y.)

**Keywords:** Pandemic preparedness, American-lineage reassortant influenza viruses, American-lineage H5N2 vaccine

## Abstract

Novel low-pathogenic avian influenza (LPAI) H5N2 viruses hit poultry farms in Taiwan in 2003, and evolved into highly pathogenic avian influenza (HPAI) viruses in 2010. These viruses are reassortant viruses containing HA and NA genes from American-lineage H5N2 and six internal genes from local H6N1 viruses. According to a serological survey, the Taiwan H5N2 viruses can cause asymptomatic infections in poultry workers. Therefore, a development of influenza H5N2 vaccines is desirable for pandemic preparation. In this study, we employed reverse genetics to generate a vaccine virus having HA and NA genes from A/Chicken/CY/A2628/2012 (E7, LPAI) and six internal genes from a Vero cell-adapted high-growth H5N1 vaccine virus (Vero-15). The reassortant H5N2 vaccine virus, E7-V15, presented high-growth efficiency in Vero cells (512 HAU, 10^7.6^ TCID_50_/mL), and passed all tests for qualification of candidate vaccine viruses. In ferret immunization, two doses of inactivated whole virus antigens (3 μg of HA protein) adjuvanted with alum could induce robust antibody response (HI titre 113.14). In conclusion, we have established reverse genetics to generate a qualified reassortant H5N2 vaccine virus for further development.

## 1. Introduction

Avian influenza (AI) viruses may not readily infect humans and cause diseases; however, mutations or reassortment give AI viruses a chance to escape the host barrier. AI H5, H6, H7, H9, and H10 subtypes are considered to have pandemic potential because of the occurrence of human infections and low level of population immunity [1]. In particular, AI H5N1 and H7N9 viruses have caused hundreds of human cases with high mortality in recent years [2]. Vaccination is a cost-effective method to mitigate impacts of an influenza pandemic and the World Health Organization (WHO) reference labs have generated many candidate vaccine viruses (CVVs) against those AI with high pandemic potentials. However, some local strains with limited circulation may not be included as a high priority but should not be ignored. 

In late 2003, low-pathogenic avian influenza (LPAI) H5N2 viruses emerged in poultry farms in Taiwan. The H5N2 viruses carry a unique gene constellation containing HA and NA genes from American-lineage H5N2 viruses and six internal genes from local H6N1 viruses. The H5N2 viruses hit poultry farms again in 2008 and evolved into highly pathogenic AI (HPAI) strains in 2010 [3,4]. A serologic surveillance conducted by Centres for Disease Control, Taiwan (Taiwan CDC) in 2012 showed that the seroconversion rate in poultry workers was 4.3%, which is significantly higher than that in general populations [5]. 

HA and NA genes of the novel H5N2 reassortant viruses belong to the American lineage which is antigenically different from the current H5N1 vaccines of the Eurasian lineage (Appendix A) [6]. Therefore, it is desirable to develop American lineage-derived influenza H5 vaccine viruses for pandemic preparedness. 

The embryonated egg-based vaccine production platform is a mature technology and has been widely used since 1945. However, the supply of the eggs may be influenced by the HPAI outbreaks. Moreover, an egg-induced antigenic drift during vaccine production may lead to poor vaccine effectiveness [7]. Therefore, alternative technology such as cell-based platforms is becoming attractive [8], including MDCK and Vero cells. Many influenza viruses grow well in MDCK cells and reach high titres with short adaptation times. Several MDCK cell-derived seasonal influenza vaccines have been licensed in Europe, South Korea, and the United States [9,10,11,12]. However, MDCK cells can only be used to produce influenza vaccines. Due to the unpredictability of influenza pandemics, multi-functional platforms are desirable for influenza pandemic preparedness [13]. Vero cells are widely used for the production of human vaccines (e.g., poliovirus, Japanese encephalitis, rabies) and could be developed as a multi-functional platform for the production of human vaccines [14]. Therefore, this study chose the Vero cell-based platform to develop H5N2 influenza vaccines for pandemic preparedness.

The current master donor virus used in the WHO reference labs, A/Puerto Rico/8/34 (PR8), was developed for the egg-based platform but not for Vero cell culture system. Therefore, a new master donor virus was required for the Vero cell-based platform. In 2010, a Vero cell-adapted high-growth H5N1 vaccine virus, Vero-15, was generated in our laboratory [15]. Vero-15 is the Vero-adapted NIBRG-14 (A/Vietnam/1194/2004 (H5N1) × PR8) virus with internal genes derived from the PR8 strain. In this study, the Vero-15 virus was further used to establish reverse genetics platforms and generate H5N2 reassortant vaccine viruses for the Vero cell culture system. 

## 2. Materials and Methods 

### 2.1. Cells, Viruses, and Medium

Vero cells were purchased from the American Type Culture Collection (ATCC, USA) and grown in M199 medium (Thermo Fisher Scientific, Waltham, MA, USA) plus 10% fetal bovine serum (FBS) (Moregate, Bulimba, Queensland, Australia). The cells were further adapted to serum-free medium, VP-SFM (Thermo Fisher Scientific, Waltham, MA, USA). For virus bulk preparation, the cells were grown in VP-SFM with Cytodex 1 microcarriers (4 g/L) (Sigma-Aldrich, St. Louis, MO, USA). When cell number reached 2 × 10^6^ cells/mL, viruses infected with 0.0001 MOI (multiplicity of infection). Avian influenza viruses, A/Chicken/CY/A2628/2012 (H5N2)—(E7, LPAI) and A/Chicken/YL/0502/2012 (H5N2)—(R3, HPAI), were provided by Animal Health Research Institute (AHRI). Vero-15 virus was the Vero cell-adapted NIBRG-14 (A/Vietnam/1194/2004 (H5N1) × PR8) virus from the National Health Research Institute (NHRI) [15]. 

### 2.2. Clone

H5N2 virus RNAs were extracted by MagNA Pure Compact Nucleic Acid Isolation Kit (Roche, Basel, Switzerland) and transcribed to cDNA with Uni12 (AGCAAAAGCAGG) primers using HiScript I Reverse Transcriptase kit (AM0670-1000, BIONOVAS, Toronto, ON, Canada). HA genes of H5N2 viruses (E7-HA: KY989967; R3-HA: KJ720208) were amplified by Bm-HA-F and Bm-NS-R, and NA genes (E7-NA: KY989968; R3-NA: KR137712) were amplified by Bm-NA-F and Bm-NA-R (Appendix A) [16]. The HA and NA genes were cloned into pHW2000 vectors [17]. Plasmids which contained HA and NA genes of E7 and R3 viruses were named as pHW-E7-HA, pHW-E7-NA, pHW-R3-HA, and pHW-R3-NA, respectively. By direct mutagenesis, the cleavage site of HA genes was modified from REKR (E7) and RRKR (R3) to GETR by primer 08-forward and 08-reverse to reduce the virus pathogenicity (Appendix A) [18], and the plasmid with modified HA gene was named pHW-E7-HAm and pHW-R3-HAm. Six internal genes of Vero-15 virus were cloned to pHW2000, which were pHW-V15-PB1, pHW-V15-PB2, pHW-V15-PA, pHW-V15-NP, pHW-V15-NS, and pHW-V15-M. 

### 2.3. Generatation of H5N2 Vaccine Viruses Using Reverse Genetics

Eight plasmids which contained NA and modified HA genes from an H5N2 virus and six internal genes from the Vero-15 virus were transfected into Vero cells to generate H5N2 vaccine viruses by electroporation (Bio-RAD Gene Pulser Xcell system, Hercules, CA, USA). A quantity of 5 × 10^6^ Vero cells were suspended in 300 μL of OPTI-MEM (Thermo Fisher Scientific, Waltham, MA, USA) and then mixed with eight plasmids (3 μg of each plasmid). The cell mixture was transferred into a 4-mm electroporation cuvette and then electroporated by two 220V square waves (20 ms duration time of each pulse and 0.1 s span between pulses). After electroporation, the cell mixture was transferred into a T25 flask which contained 5 mL pre-warmed M199 + 10% FBS and then incubated at 37 °C, 5% CO_2_ for overnight. The next day, the medium was replaced by M199 with 2 μg/mL TPCK-trypsin (Sigma-Aldrich, St. Louis, MO, USA). Transfected cells were incubated at 37 °C, 5% CO_2_, and 1μg/mL TPCK-trypsin was added to the medium every day. Generated viruses were harvested when the cells showed 90% CPE or on day 7 after the transfection. The viruses may need further passages or plaque purification in Vero cells. Virus titre was determined by haemagglutination assay (HA assay) and 50% Tissue Culture Infective Dose (TCID_50_) following the WHO standard procedures [19].

### 2.4. Sequencing 

Viral RNA was extracted by MagNA Pure Compact Nucleic Acid Isolation Kit (Roche, Basel, Switzerland), and then the extracted viral RNAs were amplified by the one-step RT-PCR (QIAGEN, Hilden, Germany). Primers for sequencing are listed in Appendix A.

### 2.5. Antigenicity Analysis

Antigenicity was analysed by using hemagglutinin inhibition assay (HI assay). Tested antigens were diluted to 4 HAU and incubated with ferret antisera, anti-R3 and anti-A/CK/CH/1209/2003 (anti-CH/2003). These two antisera were generated in the previous study to evaluate the antigenicity of H5N2 viruses isolated from 2003 to 2012 [6]. Turkey and chicken red blood cells (RBCs) were used for HI assay following WHO standard procedures [19].

### 2.6. Trypsin Dependency Test

A concentration of 1 × 10^6^ Vero cells/well was cultured in 6-well plates at 37 °C, 5% CO_2_ overnight. Before virus infection, the cells were washed with 1X PBS. Tested viruses were diluted with M199 or M199 containing 2 μg/mL TPCK-trypsin and then added to wells with a cell layer. The cells and viruses were incubated at 37 °C, 5% CO_2_ for an hour, and then washed with 1X PBS to remove unattached viruses. In a trypsin-contained group, the cells were covered with M199 containing 0.3% agarose and 2 μg/mL TPCK-trypsin, and in counterpart group, the cells were covered with M199 containing 0.3% agarose (VWR-Amersco, Radnor, PA, USA) without TPCK-trypsin. The cells were incubated at 37 °C, 5% CO_2_ for three days. The plaques were fixed in 3.7% formaldehyde and stained with 0.5% crystal violet [20].

### 2.7. Intravenous Pathogenicity Index (IVPI)

Six-week-old SPF chickens were provided by AHRI. Before injection, viruses were tested for sterilization and 10-fold diluted. Ten chickens were injected intravenously with 100 μL of diluted viruses. The chickens were examined daily for 10 days and scored [21].

### 2.8. Pathogenicity Tests and Immunization in Ferrets

Four- to six-month-old ferrets were obtained through a laboratory-breeding program at Institute of Preventive Medicine, National Defence Medical Centre. Prior to the animal study, microchips capable of measuring temperature were implanted beneath the skin between the shoulder blades. Ferrets were numbered and housed separately in animal biosafety level 3 (ABSL3) laboratories with individual temperature control at 22–24 °C and 55–60% relative humidity throughout the experiment. All study procedures and animal care were conducted in accordance with the guidelines and under the supervision of Institutional Committee on Animal Care and Use, Institute of Preventive Medicine, National Defence Medical Centre.

For the pathogenicity test, each ferret was intranasally infected with 0.5 mL of viruses, and each group had four ferrets (two males and two females). After the infection, the weight and temperature of the ferrets were monitored every day. Two ferrets (one male and one female) were sacrificed at DPI (day post-infection) 3, and the nasal turbinate, upper respiratory tract, lower respiratory tract, lung, and Hilar lymph node were collected for virus titration. Based on a pilot test, the H5N2 viruses were not pathogenic at DPI 7. Therefore, the other two ferrets were sacrificed at DPI 14 using the same procedure. 

For immunization study, four ferrets (two males and two females) were intramuscularly injected with two doses of inactivated whole virus vaccine adjuvanted with Al(OH)_3_ (Brenntag, Essen, Germany). Each dose contained 3 μg of HA protein and 300 μg of Al(OH)_3_. The ferrets were boosted 14 days after the first immunization. The sera were collected 23 days after the first immunization. 

### 2.9. Preparation of Standard Antigen and Antiserum

For standard antigen preparation, Vero cells were grown in VP-SFM with 4 g/L microcarrier using the 14-L bioreactor (CelliBen^®^ BLU Single-Use Bioreactor, International Labmate Ltd, St Albans, Hertfordshire, UK) and infected with viruses (0.0001 MOI). The supernatant of virus culture was harvested on day 4. The harvest was inactivated with 0.01% formaldehyde and purified with flow-through chromatography [22]. The purified viruses were treated with PNGase (P0704S, NEB, Ipswich, MA, USA), and the HA content was analysed by SDS-PAGE and densitometry [23]. 

Before goat immunization, the HA protein was cleaved from viruses with bromelain (Sigma-Aldrich, St. Louis, MO, USA) and purified by using sucrose gradient [24]. A goat received 20 μg of HA protein with complete Freund’s adjuvant (Sigma-Aldrich, St. Louis, MO, USA) at the first immunization. After that, the goat was boosted with 10 μg of HA protein with incomplete Freund’s adjuvant (Sigma-Aldrich, St. Louis, MO, USA) every two weeks. Until the HI titre against the given antigen, HA protein, reached 320, the serum was collected. To remove plasma protein, the serum was treated with 5% caprylic acid (Sigma-Aldrich, St. Louis, MO, USA) to the pH value of 5.5, and then centrifuged at 2400× *g* for 5 min. The supernatant was moved to a new tube and shaken at 300 rpm at 31 °C for 90 min. The supernatant was centrifuged at 3500× *g* for 45 min to remove more precipitant. The purified serum was stored at −20 °C [25].

### 2.10. Radial Immunodiffusion (SRID) Assay

Agarose gel (1%) was prepared with agarose powder (SeaKem^®^ ME Agarose, Lonza, Basel, Switzerland) and 1× PBS with 0.05% NaN_3_. The gel was mixed with the purified goat anti-E7HA serum. The ratio of serum to agarose was 1:75. The gel was placed in a circular mould and wells were generated by gel puncher. The purified viruses were treated with 1/10 V of 10% Zwittergent® 3-14 (Sigma-Aldrich, St. Louis, MO, USA) solution for 30 min and then diluted with 1× PBS with 0.05% NaN_3_. The diluted antigens were added to the wells and left overnight. After that, the gel was washed and then dried at 37 °C. Finally, the gel was stained with Coomassie blue to show the precipitin rings [26].

### 2.11. Ethics Statement

The animal experiments have followed the guidelines of Institutional Animal Care and Use Committee (IACUC) based on the Institutional Animal Care Committee Guidebook published by the US Office of Laboratory Animal Welfare under the supervision of IACUC, Animal Health Research Institutes (A02019, 2 April 2013), and IACUC, Institute of Preventive Medicine, National Defence Medical Centre (AN-102-24, 12 April 2013).

## 3. Results

### 3.1. Generation of Influenza Reassortant H5N2 Vaccine Viruses

H5N2 reassortant viruses E7-V15 and R3-V15 were generated with the NA and modified HA genes from E7 and R3 viruses and internal genes from Vero-15 virus. The viruses were passaged in Vero cells. The HA titre of E7-V15 V3 virus was 4 HAU and that of R3-V15 V3 virus was undetectable. In addition, the E7-V15 V3 also had a higher live virus titre than the R3-V15 V3 (Table 1). Therefore, E7-V15 virus was chosen for further adaptation to improve the virus yield. After 16 passages and three times of plaque purification in Vero cells, a vaccine seed virus was selected (E7-V15 C11) and its virus stock was generated (128 HAU and 3.16 × 10^7^ TCID_50_/mL). Growth efficiency of the vaccine virus was evaluated in Vero cells, which peaked to 5.62 × 10^7^ TCID/mL and 512 HAU at DPI 2 (Appendix A). 

### 3.2. Qualification of Candidate Vaccine Viruses

According to the suggestions of WHO and CDC [27,28], there are several tests for the qualification of candidate vaccine viruses (Table 2), including live virus titre, HA titre, sequence analysis, antigenic analysis, sterility test, trypsin dependency, chicken pathogenicity, ferret pathogenicity, and genetic stability of HA cleavage site. The candidate vaccine virus, E7-V15 C11, had mutations on HA (N155D, T231K, G323R, and I381V), NA (N4K) and PA (E349G) proteins compared with E7 (HA and NA) and Vero-15 (PA). The antigenicity was analysed with two ferret antisera (anti-R3 and anti-CH/2003). The HI titres of E7 and E7-V15 C11 viruses were 226 and 80 with anti-R3 sera, respectively, and 320 and 160 with anti-CH/2003 sera, respectively (Appendix A). There seemed to be a slight reduction with anti-R3 sera. Therefore, we also did the HI assay with sera from immunised ferret (Section 3.7), infected chicken (IVPI test, Section 3.4) and goat (standard reagent obtained in this study, Section 3.6). The HI titre of chicken sera against E7 and E7-V15 C11 was 113, and that of goat serum was 320. The HI titres of immunised ferret against E7-V15 C11 and E7 only had a two-fold difference (226 and 113, respectively). Overall, the HI titres of sera against E7 and E7-V15 C11 were less than four-fold different (Appendix A). Based on this result, the candidate vaccine virus has similar antigenicity to its parental virus. 

### 3.3. Genetic Stability and Trypsin Dependency of H5N2 Vaccine Viruses

To evaluate the genetic stability on HA cleavage site, E7-V15 C11 virus was passaged in Vero cells ten times and sequenced. The HA cleavage site was the same after passages (RETR↓GLF). In trypsin dependency test, E7-V15 C11 virus formed plaques when trypsin was supplied in the culture medium, but it lost the ability without trypsin (Appendix A). 

### 3.4. Intravenous Pathogenicity Index (IVPI) of H5N2 Vaccine Viruses

Nine chickens were intravenously injected with the vaccine candidate virus, E7-V15 C11, and monitored for ten days. The chickens did not show any syndromes and were alive (IVPI = 0). The sera were collected for the HI test on the 10th day after infection. The HI titres of eight chicken sera were 16 or higher (Appendix A). The seropositive rate was 89% (eight out of nine). This result shows that E7-V15 C11 virus can infect chickens, but does not cause diseases. 

### 3.5. Pathogenicity in Ferrets

Ferret model was used to assess the pathogenicity of influenza viruses in mammals. The candidate vaccine virus, E7-V15 C11, was intranasally inoculated to four ferrets (3.16 × 10^7^ TCID_50_/mL, 0.5 mL/ferret). The physical condition of the ferrets was monitored for 14 days. The body temperature of all infected ferrets was lower than 39.5 °C. The body weight change was lower than 5% (Appendix A), and the ferrets did not show any symptoms. Live virus titre of organs collected from the infected ferrets was measured with Vero cells, and there was no detectable virus titre in these samples (Appendix A). The sera from the infected ferrets were collected at DPI 14, and the homologous HI titre was higher than 20, which indicated that the candidate vaccine virus could infect ferrets with low pathogenicity.

Based on the WHO and US CDC guidelines [27,28], reassortant candidate vaccine viruses derived from H5 and H7 LPAI viruses should be assessed by monitoring the sequence of the HA cleavage site, testing the genetic stability, and conducting pathogenicity tests in ferrets and egg embryos (or chickens). The results of quality tests are listed in Table 2, which indicates that the candidate vaccine virus is lowly pathogenic and maintains a similar antigenicity with the parental virus.

### 3.6. Preparation of Standard Antigen and Antiserum

We generated standard reagents for the quantitation of HA content using E7-V15 C11 viruses. An inactivated whole virus antigen was prepared by using the serum-free microcarrier culture platform to grow viruses, and a flow-through chromatography system for purification [22]. The HA content of the whole virus antigen (H5N2 E7 Ag), as a standard antigen, was calculated as 64.8 μg/mL (36.4% of total protein) using SDS-PAGE and densitometry (Appendix A). For standard antiserum preparation, the purified virus was treated with bromelain, and the HA protein (E7 bHA) was purified using sucrose gradient. A goat was immunized with E7 bHA to generate anti-E7 bHA serum. The HI titre of the serum was 320 against H5N2 wild-type viruses, E7 and R3. Single radial immunodiffusion (SRID) assay was conducted with goat anti-E7 bHA serum and the H5N2 E7 Ag. The R-value of the SRID assay was 0.9973 (Figure 1).

### 3.7. Ferret Immunization

Four ferrets (two males and two females) were injected intramuscularly with two doses of the H5N2 E7 Ag (3 μg of each dose) adjuvanted with alum hydroxide. Interestingly, the male ferrets had a much lower serum HI titre against two wild-type H5N2 viruses compared with female ones (Table 3). Since the male and female ferrets had very different serum HI antibody responses, it is not desirable to merge them for calculating GMT. Overall, the female ferrets had similar serum HI antibody titres against two wild-type virus antigens (GMT 160 and 113) (Table 3).

## 4. Discussion

Although avian influenza viruses are considered as a risk of a future pandemic, local circulating viruses sometimes are ignored, such as the novel American-lineage H5N2 reassortant viruses. However, considering the uncertainty of disease outbreak, the potential of influenza viruses causing human infection should receive more attention. In previous studies, the H5N2 viruses were found to cause asymptomatic infection in poultry workers [5] and have different antigenicity with Eurasia-lineage H5 viruses (Appendix A) [6]. Therefore, we generated a candidate vaccine virus and related standard reagents for pandemic preparedness.

In this study, we focused on an establishment of Vero cell-based reverse genetics and production platform. Because small countries usually do not have enough space and financial support to build many factories for each kind of vaccines, a multi-functional platform is reasonable [13]. Vero cells, which are widely used for human vaccines, are suitable for this platform [14]. By using internal genes from a Vero cell-adapted high-growth H5N1 viruses, Vero-15, two Vero-derived Taiwan H5N2 reassortant viruses were generated (Table 1). 

The first step of influenza vaccine production is to generate a high-growth vaccine virus. In this study, we generated two reassortant viruses, but only one (E7-V15) had detectable HA titres. Based on the receptor preference assay with turkey and horse RBCs, the E7 virus can bind 2,3- and 2,6-sialic acid receptor, but the R3 virus could only bind 2,3-sialic acid receptor (data not shown). Therefore, the low growth efficiency of the other reassortant virus (R3-V15) may be due to the receptor preference. Overall, it is desirable to generate multiple reassortant viruses bearing different receptor binding preferences.

Because the HA and live virus titres of E7-V15 were too low for commercial production (4 HAU and 1.78 × 10^6^ TCID_50_/mL, respectively), we adapted E7-V15 in Vero cells to generate high-growth vaccine viruses. After 16 passages and three times of plaque purification in Vero cells, the HA and live virus titre of E7-V15 C11 increased to 128 HAU and 3.16 × 10^7^ TCID_50_/mL, respectively, and adapted mutations appeared on HA, NA and PA proteins. There were four adapted mutations on HA protein, N155D, T231K, G323R and I381V, and a mutation on NA protein, NA-N4K. The HA-N155D is located near a reported glycosylation site HA-154N [29]. Glycosylation at HA protein is related to antigenicity [29,30,31]. In our study, the amino acid sequences (154–158) of parental virus E7 and Vero-adapted E7-V15 C11 were N**N**VYR and N**D**VYR, respectively, so there was no glycosylation at HA-154 and HA-155 before and after adaptation. The NA mutation, NA-N4K, was located in the cytoplasmic tail of NA protein (M NP**N**QK). The cytoplasmic tail of NA protein is important for lipid raft targeting, virus assembly and morphology [32,33,34,35,36], not for antigenicity. Besides, our study shows that the antigenicity of E7 and E7-V15 C11 were similar based on serum HI assay (Appendix A). Therefore, Vero-adapted mutations on HA and NA proteins in this study seemed not to change the antigenicity of E7-V15 C11 compared with E7.

SRID is the standard assay for quantification of HA contents [37]. However, the preparation of standard reagents (antigen and antiserum) for SRID is time-consuming and labour-intensive, which could delay the supply of pandemic influenza vaccines. Currently, only four essential regulatory laboratories (ERLs) are invited to generate these standard reagents, and it will be desirable to include more laboratories to accelerate the preparation of standard reagents for pandemic preparedness [38]. In this study, we successfully generated the standard reagents for SRID to quantify our vaccine antigen for the ferret study.

Ferret is the standard animal model for evaluating human influenza vaccines [28]. We assessed the pathogenicity and immunogenicity of the H5N2 vaccine in ferrets, and found that the H5N2 candidate vaccine virus was lowly pathogenic and could elicit a robust antibody response in female ferrets but not male ferrets (Table 3). We found three previous ferret vaccination studies which evaluated H5 viruses (Eurasian-lineage) and commercial adjuvants, and provided serum antibody response and gender information (two female studies and one male study) [39,40,41]. In the female ferret studies, ferrets were immunised with 3.8 μg of split antigens and oil-in-water adjuvant (GSK), and robust immune responses were observed (serum HI antibody ~100–300), which is similar to the finding in our study (Table 3). In the male study, ferrets were immunised with 7.5 μg of split antigens and oil-in-water adjuvants (AS03), strong immune responses were observed (serum HI antibody about 400). It is hard to make a conclusion about gender difference in immune response to influenza vaccination as different antigen types and adjuvants used in different studies. 

Influenza risk assessment tool (IRAT) helps the decision of vaccine development [42,43]. For lowly pathogenic influenza viruses, generation of high-growth candidate vaccine viruses may be enough. When the risk increases, it may be necessary to initiate vaccine production and clinical trials. In this study, a high-growth candidate vaccine virus has been generated using reverse genetics for a Vero cell-based production platform. Once the risk of the H5N2 viruses increases, the candidate vaccine virus can be used for production and start of clinical trials.

## Figures and Tables

**Figure 1 viruses-11-00543-f001:**
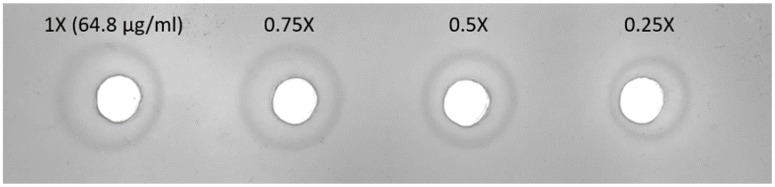
Single radial immunodiffusion (SRID) assay of goat anti-E7 bHA serum against the H5N2 E7 Ag.

**Table 1 viruses-11-00543-t001:** The profile of generated H5N2 reassortant viruses.

Strain ID	E7-V15 V3	R3-V15 V3
Source of HA and NA genes	A/CK/CY/A2628/2012 (E7, LPAI)	A/CK/YL/0502/2012 (R3, HPAI)
Modified cleavage site	GGA GAA ACA AGA (GETR)	GGA GAA ACA AGA (GETR)
Source of internal genes	Vero-15	Vero-15
HA titre	4 HAU	<2 HAU
Virus titre	1.78 × 10^6^ TCID_50_/mL	3.98 × 10^5^ TCID_50_/mL

**Table 2 viruses-11-00543-t002:** The summary of vaccine quality tests of E7-V15 C11 virus.

Test Items	Result
Live virus titre	3.16 × 10^7^ TCID_50_/mL (5.44 × 10^7^ PFU/mL)
HA titre	128 HAU
Sequence analysis ^1^	HA (N155D, T231K, G323R, I381V) NA (N4K) PA (E349G)
Antigenic analysis	Similar to parental strain (E7)
Sterility test	No bacteria contamination
Trypsin-dependency	Trypsin dependent
Chicken pathogenicity	IVPI = 0
Ferret pathogenicity	Low pathogenicity (weight loss <5%)
Genetic stability of HA cleavage site	Without mutations on the cleavage site after 10 passages

^1^ The HA and NA genes were compared with HA and NA of E7. The internal genes were compared with those of Vero-15.

**Table 3 viruses-11-00543-t003:** The HI titre of ferret sera after two intramuscular injections of H5N2 inactivated virus vaccine.

Vaccine Strain	E7-V15 C11+ Al(OH)_3_				
Ferret no.	259 (M)	269 (M)	265 (F)	266 (F)	GMT (M/F)
YL/120502 (2012)- R3	<10	10	320	80	7/160
CY/A2628 (2012)- E7	<10	10	160	80	7/113

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
