# Peer review of "Development of American-Lineage Influenza H5N2 Reassortant Vaccine Viruses for Pandemic Preparedness"

_viruses, 2019, doi:10.3390/v11060543_

Reviewer 1 Report

General comments:

This manuscript describes creation and analysis of the candidate vaccine H5N2 virus strain. The vaccine is intended to provide a basis for rapid assessment and production should the need for a H5N2 vaccine arise in the future. Currently there is no vaccine candidate available for the H5N2 that evolved in Taiwan, presumably through reassortment of the American-lineage H5N2 HA and NA genes and six internal genes from H6N1 virus and subsequent acquisition of the highly pathogenic phenotype around 2010. The presented candidate vaccine virus was generated and produced in Vero cells that are not currently used for influenza vaccine production but could be utilized in case of a pandemic. Pathogenesis and antigenicity of the recombinant vaccine virus was examined in Ferrets – a current gold standard for the animal model of influenza.

The study is methodical and straightforward. It could be suitable for publication in Viruses if authors presented evidence of protection in animal model of infection, even mouse, and demonstrated that the protection offered by this vaccine is superior to the currently existing H5N1 vaccines.

Major comments:
1) No infection studies in mice, ferrets, or chickens that would demonstrate protection from H5N2 infection (even LPAI) offered by this new vaccine is presented. Vaccine clearly induces specific antibody responses, but it is unclear how whether these responses are sufficient and broad enough to offer protection from the circulating H5N2 strains.

2) Insufficient evidence is presented in regard to antigenic differences between American-lineage H5 and Eurasian-lineage H5 and how these differences affect effectiveness of candidate H5N1 vaccines against H5N2 viruses that circulate in Taiwan.

3) Authors list a number of mutations that arose during Vero cell production. But potential significance of these mutations was not discussed. Especially relevant are 4 mutations in HA and 1 in NA. How these mutations affect antigenicity?

Author Response

Major comments:
1) No infection studies in mice, ferrets, or chickens that would demonstrate protection from H5N2 infection (even LPAI) offered by this new vaccine is presented. Vaccine clearly induces specific antibody responses, but it is unclear how whether these responses are sufficient and broad enough to offer protection from the circulating H5N2 strains.

Response: Ferrets are the most relevant animal model for evaluating the immunogenicity and protection of influenza vaccines for human use. Because the highly-pathogenic avian influenza H5N2 viruses are low or no pathogenic in ferrets (R3 WT, Supplemental Figure S3 and Table S5), it is not desirable to evaluate the protection of the H5N2 vaccine in ferrets. Besides, the HI antibody response in ferret could correlate the antibody response in human. Therefore, we only evaluate the antibody response in ferrets in this study. (Chia MY et al., 2015, PLoS ONE and Wu UI et al., 2017, Vaccine)

2) Insufficient evidence is presented in regard to antigenic differences between American-lineage H5 and Eurasian-lineage H5 and how these differences affect effectiveness of candidate H5N1 vaccines against H5N2 viruses that circulate in Taiwan.

Response: The manuscript was revised as suggested, and the data has been added to the supplementary material (Table S1) and mentioned in the manuscript (Introduction, line 57). The result shows that the Eurasian-lineage viruses are antigenically different from American-lineage viruses. Therefore, it is desirable to generate vaccine viruses for both lineages for pandemic preparedness.

3) Authors list a number of mutations that arose during Vero cell production. But potential significance of these mutations was not discussed. Especially relevant are 4 mutations in HA and 1 in NA. How these mutations affect antigenicity?

Response: Thanks for the suggestions. The discussion of Vero-adapted mutation on HA and NA has been added to the discussion (line 310~320). Based on the result of the serological assay, E7-V15 C11 has similar antigenicity with E7, even there are mutations on HA and NA proteins (Table S3).

Additional Reference:

Chia M-Y, Hu AY-C, Tseng Y-F, Weng T-C, Lai C-C, Lin J-Y, et al. (2015) Evaluation of MDCK Cell-Derived Influenza H7N9 Vaccine Candidates in Ferrets. PLoS ONE 10(3): e0120793. doi:10.1371/ journal.pone.0120793

U.-I. Wu et al./Vaccine 35 (2017) 4099–4104

Reviewer 2 Report

The manuscript by Chen at al. describes the development and characterization of a candidate influenza vaccine virus adapted to grow in Vero cells for protection against a potentially zoonotic H5N2 avian influenza that has been circulating in Taiwan’s poultry for over a decade. The candidate vaccine is evaluated for several features as recommended by the guidelines from the US CDC and the WHO.  

This work is interesting because current influenza vaccines are far from perfect and new vaccine production strategies are needed in preparation for the next pandemics, for the eventuality of egg shortages due to substantial avian mortality and for many other reasons. The manuscript could be more informative by providing comparative numbers between the performance of this vaccine candidate and the conventional vaccines. Some statements regarding the pandemic risk of the H5N2 virus, or the antigenicity and stability of the rescued virus should be toned-down.

Specific comments:

Line 62 (Introduction). The authors claim that egg adaptation can result in antigenic changes. While this is correct, it is probably more relevant in the case of mammalian-adapted influenza viruses (such as the seasonal influenzas) that need to adapt to the use of avian receptors. On the other hand it can be equally argued that an avian virus (such as the H5N2 used in this study) might undergo antigenic changes during adaptation to grow in mammalian cells (such as the Vero cells used in this study). The authors should consider discussing this.

Line 76 (Introduction). For clarity, the authors should mention that the six internal segments of the virus H5N1 Vero-15 are derived from the PR8 strain, not from an avian H5N1 virus.

Line 80 (Material and Methods). Chicken embryo fibroblasts (CEF) should be mentioned in this paragraph, since they were used in the titration of the virus from ferret tissues (line 235).

Line 198. Passage in Vero cells resulted in improved replication of the viruses generated by reverse genetics. The exact adaptation process (how many passages, how many plaque purifications) should be described. For comparison, it would be useful to mention what are the viral titres obtained routinely with a conventional vaccine virus in embryonated eggs.

Line 212. The Vero-adapted virus acquired a remarkable number of amino acid changes, mostly in the immunodominant HA protein. Antigenic characterization is limited to comparison of HAI titers by 2 ferret antisera that indeed show a slight reduction in HA inhibition of the Vero-adapted virus by both antisera. Nevertheless, the authors conclude that both viruses have similar antigenicity. This conclusion would be better supported with a larger panel of antisera, including the (female) ferret sera obtained after immunization with the inactivated vaccine candidate (line 258) and the goat serum obtained as standard reagent (line 244).

Line  236. In the supplemental table S4 it is shown that the wild type virus R3 resulted in significantly higher antibody titres than the vaccine candidate, in spite of using a much lower dose (almost 100 times lower). Also, in the ferret immunization experiment, using two doses of adjuvanted, inactivated virus, the two male ferrets had barely detectable antibody titres. This suggests low antigenicity of the vaccine candidate virus which is concerning and should be discussed.  For comparison it would also be useful to know what are the HAI titres expected from a conventional vaccine virus used in the same conditions.

Line 248. It is not clear which virus was used to prepare the standard antigen: was it the parental H5N2 E7 or the Vero-adapted E7-V15 C11? The text says H5N2 E7 Ag (line 253) but the figure legend says purified E7-V15 C11 (line 257).

Line 265 (Table 3). What does “entry 2” stands for?

Author Response

Specific comments:

Line 62 (Introduction). The authors claim that egg adaptation can result in antigenic changes. While this is correct, it is probably more relevant in the case of mammalian-adapted influenza viruses (such as the seasonal influenzas) that need to adapt to the use of avian receptors. On the other hand it can be equally argued that an avian virus (such as the H5N2 used in this study) might undergo antigenic changes during adaptation to grow in mammalian cells (such as the Vero cells used in this study). The authors should consider discussing this.

Response: Thanks for the suggestions. The discussion of Vero-adapted mutation on HA and NA has been added to the discussion (line 310~320). Based on the result of the serological assay, E7-V15 C11 has similar antigenicity with E7, even there are mutations on HA and NA proteins (Table S3).

Line 76 (Introduction). For clarity, the authors should mention that the six internal segments of the virus H5N1 Vero-15 are derived from the PR8 strain, not from an avian H5N1 virus.

Response: Thank you for the suggestion. Vero-15 is the Vero-adapted NIBRG-14 viruses (A/Vietnam/1194/2004 (H5N1) x PR8), so the internal genes are derived from PR8 strain. This information has been added in the introduction (line 76~78).

Line 80 (Material and Methods). Chicken embryo fibroblasts (CEF) should be mentioned in this paragraph, since they were used in the titration of the virus from ferret tissues (line 235).

Response: The manuscript was revised as suggested. The cultivation of chicken embryo fibroblasts (CEF) has added to material and methods (line 91~101).

Line 198. Passage in Vero cells resulted in improved replication of the viruses generated by reverse genetics. The exact adaptation process (how many passages, how many plaque purifications) should be described. For comparison, it would be useful to mention what are the viral titres obtained routinely with a conventional vaccine virus in embryonated eggs.

Response: The Vero-adapted H5N2 viruses (E7-V15 C11) was generated through 16 passages and 3 times of plaque purification in Vero cells (this information has been added to (Result 3.1, line 209~210). The conventional vaccine viruses in eggs usually 512~2048 HAU, but in Vero cells usually about 128~256 HAU (Kistner, O. et al., 1998, Vaccine). Therefore, we adapted the H5N2 vaccine virus until the HA titre reached 128 HAU (line 211).

Line 212. The Vero-adapted virus acquired a remarkable number of amino acid changes, mostly in the immunodominant HA protein. Antigenic characterization is limited to comparison of HAI titers by 2 ferret antisera that indeed show a slight reduction in HA inhibition of the Vero-adapted virus by both antisera. Nevertheless, the authors conclude that both viruses have similar antigenicity. This conclusion would be better supported with a larger panel of antisera, including the (female) ferret sera obtained after immunization with the inactivated vaccine candidate (line 258) and the goat serum obtained as standard reagent (line 244).

Response: Thanks for the suggestion. We did the HI assay with sera from immunised ferret (female), infected chicken (IVPI test) and goat (standard reagent obtained in this study). The result has been added to line 223~230. Overall, the HI titres of sera against E7 and E7-V15 C11 were less than 4-fold differences (Table S3). Based on this result, the candidate vaccine virus has similar antigenicity with its parental virus.

Line 236. In the supplemental table S4 it is shown that the wild type virus R3 resulted in significantly higher antibody titres than the vaccine candidate, in spite of using a much lower dose (almost 100 times lower).

Response: We did consider using both E7 and R3 to generate H5N2 vaccine seed viruses. Because of the undetectable HA titre of R3-V15 virus, we chose E7-V15 virus for further adaptation (Result 3.1, line 205~209).

Also, in the ferret immunization experiment, using two doses of adjuvanted, inactivated virus, the two male ferrets had barely detectable antibody titres. This suggests low antigenicity of the vaccine candidate virus which is concerning and should be discussed. For comparison it would also be useful to know what are the HAI titres expected from a conventional vaccine virus used in the same conditions.

Response: The manuscript was revised as suggested. We tried to compare our finding with other studies but it is hard to make a conclusion about gender difference in immune response to influenza vaccination as different antigen types and adjuvants used in different studies. This discussion has been added to line 331~339.

Line 248. It is not clear which virus was used to prepare the standard antigen: was it the parental H5N2 E7 or the Vero-adapted E7-V15 C11? The text says H5N2 E7 Ag (line 253) but the figure legend says purified E7-V15 C11 (line 257).

The virus used to prepare standard antigen is Vero-adapted E7-V15 C11, and the bHA used to produce standard serum is also derived from the Vero-adapted E7-V15 C11 virus. We just want to simplify the name of standard reagents. This part has been clarified in the manuscript (Result 3.6, line 263).

Line 265 (Table 3). What does “entry 2” stands for?

Thanks for your mention. It is a typo mistake. I pasted my article to the example of format and missed it. It is “CY/A2628 (2012)- E7” and revised in the manuscript (Table 3, line 284).

Additional reference:

Kistner, O.; Barrett, P. N.; Mundt, W.; Reiter, M.; Schober-Bendixen, S.; Dorner, F., Development of a mammalian cell (Vero) derived candidate influenza virus vaccine. Vaccine 1998, 16, (9-10), 960-8.

Round  2

Reviewer 1 Report

Authors addressed this reviewer's concerns adequately.

To be acceptable for publication, the tables S1 (new) and S5 that contain previously published information (Virology 508 (2017) 159-163) have to be modified with all previously published information removed and replaced with a comparative description in the legend with reference to Virology 508 (2017) 159-163. Or, each numerical value taken from other publication should have the reference number explicitly present next to it in parenthesis or as superscript font and clearly separated from new data using font modification (e.g. italics).

Author Response

To be acceptable for publication, the tables S1 (new) and S5 that contain previously published information (Virology 508 (2017) 159-163) have to be modified with all previously published information removed and replaced with a comparative description in the legend with reference to Virology 508 (2017) 159-163. Or, each numerical value taken from other publication should have the reference number explicitly present next to it in parenthesis or as superscript font and clearly separated from new data using font modification (e.g. italics)

Response: The manuscript has been revised as suggested. The previously published information (data of R3 infected ferrets) has been removed from Figure S3 and Table S1 and Table S5.
